# Short noncoding RNAs as predictive biomarkers for the development from inflammatory bowel disease unclassified to Crohn's disease or ulcerative colitis

Jaslin P. James[1]*, Lene Buhl Riis[1,2], Rolf Søkilde[3], Mikkel Malham[4,5], Estrid Høgdall[1,2], Ebbe Langholz[2,6], Boye Schnack Nielsen[3]

1 Department of Pathology, Herlev University Hospital, Herlev, Denmark, 2 Institute of Clinical Medicine, University of Copenhagen, Copenhagen, Denmark, 3 Bioneer A/S, Hørsholm, Kogle Allé 2, Hørsholm, Denmark, 4 The Pediatric Department, Copenhagen University Hospital—Amager and Hvidovre, Hvidovre, Denmark, 5 Copenhagen Center for Inflammatory Bowel Disease in Children, Adolescents and Adults, Hvidovre Hospital, University of Copenhagen, Hvidovre, Denmark, 6 Gastroenheden, Herlev University Hospital, Herlev, Denmark

* jaslin.pallikkunnath.james@regionh.dk

**Data Availability Statement:** The raw data generated during and/or analysed during the current study are not publicly available, which is in

## Abstract

Numerous pathogenic processes are mediated by short noncoding RNAs (sncRNA). Twenty percent of inflammatory bowel disease (IBD) patients are labelled as IBD unclassified (IBDU) at disease onset. Most IBDU patients are reclassified as Crohn's disease (CD) or ulcerative colitis (UC) within few years. Since the therapeutic methods for CD and UC differ, biomarkers that can forecast the categorization of IBDU into CD or UC are highly desired. Here, we investigated whether sncRNAs can predict CD or UC among IBDU patients. 35 IBDU patients who were initially diagnosed with IBDU were included in this retrospective investigation; of them, 12, 15, and 8 were reclassified into CD (IBDU-CD), UC (IBDU-UC), or remained as IBDU (IBDU-IBDU), respectively. Eight IBD patients, were included as references. SncRNA profiling on RNA from mucosal biopsies were performed using Affymetrix miRNA 4.0 array. Selected probe sets were validated using RT-qPCR. Among all patients and only adults, 306 and 499 probe sets respectively were differentially expressed between IBDU-CD and IBDU-UC. Six of the probe sets were evaluated by RT-qPCR, of which miR-182-5p, miR-451a and ENSG00000239080 (snoU13) together with age and sex resulted in an AUC of 78.6% (95% CI: 60–97) in discriminating IBDU-CD from IBDU-UC. Based on the three sncRNAs profile it is possible to predict if IBDU patients within 3 years will be reclassified as CD or UC. We showed that the expression profile of IBDU patients differ from that of definite CD or UC, suggesting that a subgroup of IBDU patients may compose a third unique IBD subtype.

## Introduction

Inflammatory bowel disease unclassified (IBDU) is a subtype of inflammatory bowel disease (IBD) alongside ulcerative colitis (UC) and Crohn's disease (CD). IBDU is characterised by

accordance with the rules concerning processing of personal data set out in the EU General Data Protection Regulation (GDPR) and the Danish Data Protection Act. Nonetheless, may a researcher have an interest in our data they are welcome to contact us and collaborate. The data that support the findings of this study can be requested from The National Secretariat for Bio- and Genome Bank Denmark, RBGB.sekretariat.herlev-og-gentofte-hospital@regionh.dk, Herlev Hospital, Borgmester Ib Juuls Vej 73, 2730 Herlev, Denmark. Normalized data of the gene expression from the discovery study is uploaded to gene expression omnibus with the reference number: GSE228622.

**Funding:** This work was supported by Aage Louis Hansen fonden [J.nr.20-2B-5995]; Herlev internal research funding [after year, 2019]; Colitis-Crohn Foreningen [2021]; Torben og Alice Frimodts Fond [FK1900]; Aase and Ejnar Danielsen's Foundation [20-10-0411] All the funding was received towards JPJ. The funders had no role in study design, data collection and analysis, decision to publish, or preparation of the manuscript.

**Competing interests:** The authors have declared that no competing interests exist.

clinical and endoscopic symptoms of chronic colitis without clear distinct characteristics of UC or CD, but rather modest features of both [1]. Differential diagnosis is crucial for appropriate clinical management of IBD patients since each subtype of IBD entails unique therapeutic strategies [2,3]. Based on a number of epidemiological studies, Tontini estimated that 10% of IBD patients in Europe and North America are diagnosed with IBDU [4]. Among paediatric IBDU patients, IBDU is twice as prevalent as adult-onset IBDU, with younger ages having the highest prevalence [5]. Up to 80% of paediatric IBDU patients will later be diagnosed with either CD or UC [5,6]. This suggests that some IBDU patients may represent early manifestations of CD or UC [7]. Interestingly, another study has shown up to 69% of paediatric IBDU patients remain diagnosed as IBDU into their adulthood [8]. The dynamics and the eluding character of IBDU suggest that a specific diagnosis at disease onset is a moving target, thus preventing identification of strong molecular profiles for differential diagnosis of IBDU patients.

We and others have previously reported on genetic and epigenetic biomarkers for differentiating CD and UC samples based on mucosal tissue biopsies [4,9,10]. In line with findings regarding gene expression in inflamed mucosa, IBD genetics have emphasized a shared T helper (Th) 17/Interleukin (IL) 23 pathway for CD and UC, and showed relatively few disease-specific mechanisms, such as barrier integrity relating to UC and autophagy relating to CD [11–14]. This level of genetic risk sharing suggests that nearly all the susceptibility loci associated with IBD may also have an impact on both IBD subtypes, which limits the use of genetic panels to resolve discrimination of CD and UC. The majority of the single nucleotide polymorphisms (SNPs) associated with IBD are found in noncoding areas of the genome, suggesting an indirect function in the control of gene expression [15]. The early epigenetic studies in IBD identified differential expression of small noncoding RNAs such as microRNAs (miRNAs) in UC colonic mucosa compared to normal mucosa [16]. Combinations of epigenetic alterations may result in activation of genes that promote chronic inflammation or of blocking anti-inflammatory genes [17]. Certain mRNA levels have been demonstrated to be closely controlled by miRNAs resulting in significant changes in protein synthesis [18], for example Let-7e and Let-7f mediated dysregulation of IL-23R signalling linked to an SNP in the IL-23R gene that is associated with IBD susceptibility [19]. E.g. long noncoding RNAs, miRNAs, and circular RNAs have been studied both as biomarkers and as therapeutic targets in IBD [20,21]. MiRNAs and snoRNAs constitute two main classes of short noncoding RNAs (sncRNAs). In contrast to miRNAs that contain 18–22 nucleotides, snoRNAs have 60 to 300 nucleotides. Many miRNAs mediate post-transcriptional gene silencing by controlling translation [22,23], whereas the majority of snoRNAs function as guide RNAs at the post-transcriptional level by modification of nucleotides in various RNAs, including mRNAs [24]. We recently reviewed microRNAs as prognostic, diagnostic, and predictive biomarkers in IBD [9] and found that several studies have explored the potential of blood, tissue or saliva-derived miRNAs as biomarkers for differentiating CD from UC. SnoRNAs have been found dysregulated in inflammation [25,26], and in one study, snoRNAs were found to be involved in the progression of UC [27].

In this study we assembled endoscopy biopsies from adults and paediatric patients diagnosed with IBDU. 36% of the IBDU cases were reclassified as CD (IBDU-CD), 42% of the IBDU cases as UC (IBDU-UC) and 22% remained as IBDU (IBDU-IBDU). We explored the predictive value of sncRNAs, in the preliminary diagnostic workup in IBDU patients, and report a three-gene-target RT-qPCR that can help in the prediction of classification of IBDU patients.

## Materials and methods

This study was a retrospective study examining differential expression of sncRNAs in mucosal tissue biopsies collected at time of IBD diagnosis as IBDU.

## Ethical considerations

The study has been approved by the National Ethical Committee (j. nr. H-20032221) and the Danish Data Protection Agency (P-2020-737). The study was conducted according to the declaration of Helsinki, in accordance with the ethical standards and according to national and international guidelines.

## Samples

Archived formalin-fixed paraffin-embedded (FFPE) mucosal tissue biopsies from 35 IBDU patients and 8 IBD patients were used in the discovery study. Multiple biopsies from the affected parts of the colon or rectum were collected by colonoscopy performed as part of the diagnostic work-up and stored in clinical biobanks at hospitals in the capital region of Denmark. One biopsy per patient was chosen based on the availability in the biobank and uniform location as possible from the affected tissue. Based on the patient charts, the samples from IBDU patients were reclassified into CDs or UCs or remained as IBDU after minimum 3 years of follow up, i.e. 12 IBDU patients were reclassified as CD (IBDU-CD) and 15 IBDU patients as UC (IBDU-UC). Eight IBDU patients remained as IBDU until 2022. All IBDU samples had thorough clinical evaluation including gastroduodenoscopy, colonoscopy, and small bowel MRI at the time of diagnosis in order to significantly reduce the risk of misclassification. Diagnosis of IBDU was based on Porto criteria [28] (children) or Copenhagen criteria (adults [29,30]). The FFPE blocks used in this study were collected from 1998 to 2018, and the study was conducted in 2022 to 2023. Only LBR, MM and EL had access to the patient files during the study. All patients included in the study were followed at Department of Gastroenterology, Copenhagen University Hospital—Herlev and Gentofte, Denmark or the Department of Paediatric and Adolescence Medicine, Copenhagen University Hospital—Hvidovre, Denmark. According to the amount of total RNA remaining after the discovery study, 32 IBDU samples (11 IBDU-CDs, 14 IBDU-UCs and 7 IBDU-IBDUs) were available for RT-qPCR validation.

## RNA extraction

Total RNA was extracted from two 10μm thick sections of FFPE tissue samples using MagMAX™ FFPE DNA/RNA Ultra Kit from Applied Biosystems™ for the microarray discovery study and MiRNeasy FFPE Kit from Qiagen for the RT-qPCR validation study. Both RNA extraction protocols were performed according to the manufacturer's instructions. RNA quantity and purity were measured using the NanoDrop spectrophotometer (ThermoFisher Scientific) and Qubit RNA HS Assay kit (Life Technologies). Total RNA samples were stored at −80°C until further use.

## MiRNA microarray

Expression levels of miRNAs in the discovery study samples were assessed using GeneChip® microarray and Flashtag™ Bundle (Affymetrix, CA, USA). Labeling for miRNAs with FlashTag Biotin HSR was performed with samples containing 130ng of total RNA in 8μl of nuclease free water according to the manufacturer's instructions. Samples were then hybridized to GeneChip™ miRNA 4.0 microarrays. Afterwards the miRNA microarray chips were washed and stained using the Genechip Fluidics Station 450. Subsequently miRNA microarray chips were scanned using an Affymetrix GeneChip 7G scanner.

## RT-qPCR

**Primer design.**   Sequences for RT-qPCR primers for miR-182-5p, miR-451a, miR-628-5p, miR-1298-3p, miR-4793-3p and snoU13 were obtained from miRbase (https://www.mirbase.org/

) for miRNAs and Ensembl (http://www.ensembl.org/) for the snoRNA. Forward and reverse primers were designed using miRprimer software [31]. Primer3 version 4.1.0 (https://primer3.ut.ee/) was used for the design of primers for snoRNAs and RN7SL, with default settings and a Product Size Range of 40-200bp. The primers used in the custom panel are listed in S1 File. Primers were purchased from IDT as standard small-scale DNA oligos with desalting (IDTDNA).

**cDNA synthesis and qPCR reaction.** cDNA synthesis was performed as described in Balcells et al. [32], with the following reagents: 10ng of total RNA, Poly-A polymerase (NEB: M0276L), MulV RT enzyme (NEB M0253L) and dNTPs (NEB N0447L). The cDNA was diluted 10-fold for the RT-qPCR. The qPCR reaction was performed in RealQ Plus 2x Master Mix Green (without Rox, Ampliqon, A323406), with cycling conditions: 1x 95˚C for 15 minutes, 40 cycles of (95˚C for 30sec and 60˚C for 30sec) using a Light Cycler 480 system (Roche).

## Statistical analysis

**Microarray data analysis.** Original datafiles were retrieved from the command console as.CEL files and further data analysis was performed in RStudio with R version 4.2.2. MiRNA expression among different samples were preprocessed using robust multichip average (RMA) algorithm, including background subtraction, quantile normalization and summarization [33,34]. Lowly expressed probe sets were removed, and batch correction was performed using ComBat batch correction [35]. Probe sets were filtered for species homo sapiens. For annotation, pd.mirna.4.0 package [36] was used. Subsequently linear modeling was performed to find the differentially expressed miRNAs between IBDU-CDs and IBDU-UCs using limma package [37]. Probe sets were sorted according to the Benjamini-Hochberg adjusted-p-values (adjusted for multiple testing) and fold changes in expression values. The AUC (area under the receiver operating characteristic curve (ROC)) was calculated as a measure of discrimination, and the adjusted p values less than 0.05 were considered significant. Multivariate analysis including age, sex, disease duration, and sample location were performed. Significant probe sets were filtered for MirGeneDB 2.1 database [38,39] for further RT-qPCR based validation, except miR-4793-3p and ENSG00000239080 (snoU13). Those two probe sets were included for further RT-PCR validation as miR-4793-3p was already a common potential candidate from the discovery study in both analysis 1 and 2, and snoU13 had AUC values > 80% for both transcript forms. Microarray expression data is uploaded into the gene expression omnibus with series name GSE228622.

MiRNA differential expression from microarray data were analyzed as described above in two different set of samples, such as analysis 1 with **all** IBDU samples (12 IBDU-CDs, 15 IBDU-UCs, and 8 IBDU-IBDUs) and analysis 2 with **only adult** IBDU (9 IBDU-CDs, 8 IBDU-UCs, and 5 IBDU-IBDUs) samples. Analysis with only paediatric samples was not performed, due to the limited number of samples in this group.

**RT-qPCR data analysis.** Melting curve analysis and late Cq values were used to filter for assays with missing detection. This was the case for miR-628-5p, miR-1298-3p and miR-4793-3p, which were below detection limit and hence they were excluded from further analysis. RT-qPCR data was analyzed using the $2^{-(\Delta Ct)}$ method [40]. Primer pairs for miR-27a-3p, miR-16-5p, miR-191-5p, SNORD48 and RN7SL were used as reference assays according to previous literatures and manufacture's recommendation [41–43]. The geometric mean of all the reference probe sets were used to yield relative fold expression levels of the 7 different probe sets [44]. All statistical analysis for RT-qPCR data were performed using R. AUC values for discriminating CD from UC, using the selected sncRNAs expression values from RT-qPCR were calculated for all samples. Different generalized linear models (GLM) were created using GLM function in R and different combinations of factors such as age at the time of diagnosis, sex,

and relative expression values of miR-182-5p, miR-451a, and/or snoU13. To identify which of the miRNAs could discriminate IBDU-CD from IBDU-UC, we trained the GLM models in the RT-qPCR based expression data. The AUC method was used to summarize the prediction's sensitivity and specificity. Genes which have AUC values close to 1 (100%) are more effective at discriminating between CD and UC patients.

## Results

A total of 43 archived mucosal biopsies from IBD patients are included and the patient characteristics are summarized in Table 1. Note that 35 patients (13 paediatric patients and 22 adult patients) were diagnosed as IBDU and later reclassified into either CD (n = 12), UC(n = 15), or remained as IBDU (n = 8). A graphical representation of the sample collection and workflow of the study is shown in Fig 1.

### Discovery study

Microarray analysis was performed on RNA isolated from endoscopic pinch biopsies of the whole sample cohort. 306 probe sets were differentially expressed between IBDU-CD and IBDU-UC (Fig 2A). We found six probe sets, including miR-182-5p, miR-3176, miR-483-5p, miR-4462, miR-4793-3p, and miR-668-5p with differences of more than one log fold change in expression between IBDU-CD and IBDU-UC. Expression of miR-4462, miR-483-5p, miR-4793-3p, and miR-668-5p was found to be lower ($P<0.05$), whereas expression of miR-182-5p and miR-3176 was found to be higher in IBDU-CD compared to IBDU-UC. Fig 2B shows a heat map presenting the expression values of the six probe sets. The IBDU-CD and IBDU-UC

**Table 1. Patient characteristics of the discovery study and data validation cohort (subset of discovery cohort).**

| | Discovery Study cohort (N = 43) | | | | | Data validation cohort (subset of discovery cohort) (N = 32) | | |
| --- | --- | --- | --- | --- | --- | --- | --- | --- |
| | IBDU | | | IBD | | IBDU | | |
| | **CD** | **UC** | **IBDU** | **CD** | **UC** | **CD** | **UC** | **IBDU** |
| Number of samples | 12 | 15 | 8 | 4 | 4 | 11 | 14 | 7 |
| Age at diagnosis | | | | | | | | |
| Median (IQR) | 36 (18–59) | 17 (12–30) | 26 (13–41) | 32 (27–41) | 39 (37–48) | 24 (15–59) | 20 (13–35) | 35 (14–51) |
| Sex | | | | | | | | |
| Female, N (%) | 6 (50) | 6 (40) | 2 (25) | 2 (50) | 2 (50) | 5 (45) | 6 (43) | 2 (29) |
| Age of the FFPE sample blocks (in years) | | | | | | | | |
| Median (IQR) | 10 (9–13) | 16 (7–19) | 16(10–18) | 8 (7–9) | 2 (2–2) | 10 (8–13) | 15 (6–20) | 16 (9–18) |
| Disease duration as IBDU (in years) | | | | | | | | |
| Median (IQR) | 8 (7–9) | 9 (5–16) | 16 (9–18) * | NAP | NAP | 7 (6–10) | 9 (4–17) | 15 (7–18) |
| Number of samples and sample location | | | | | | | | |
| • Colon (unspecified) | 3 | 4 | 4 | | | 3 | 4 | 3 |
| • Ascending colon | | 1 | | | | | 1 | |
| • Transverse colon | | 1 | | | | | 1 | |
| • Descending colon | 2 | 2 | 1 | 2 | 1 | 3 | 1 | 1 |
| • Sigmoid colon | 6 | 5 | 4 | 2 | 1 | 5 | 4 | 3 |
| • Rectum | 1 | 3 | | | 2 | | 3 | |

N-Number of samples, IBDU-Inflammatory Bowel Disease Unclassified, CD- Crohn's disease, UC- Ulcerative colitis, HC- Healthy controls, NAP-Not applicable, IQR-Inter quartile range, FFPE- Formalin fixed paraffin embedded

*—Number of years where IBDU samples remained as IBDU until 2022.

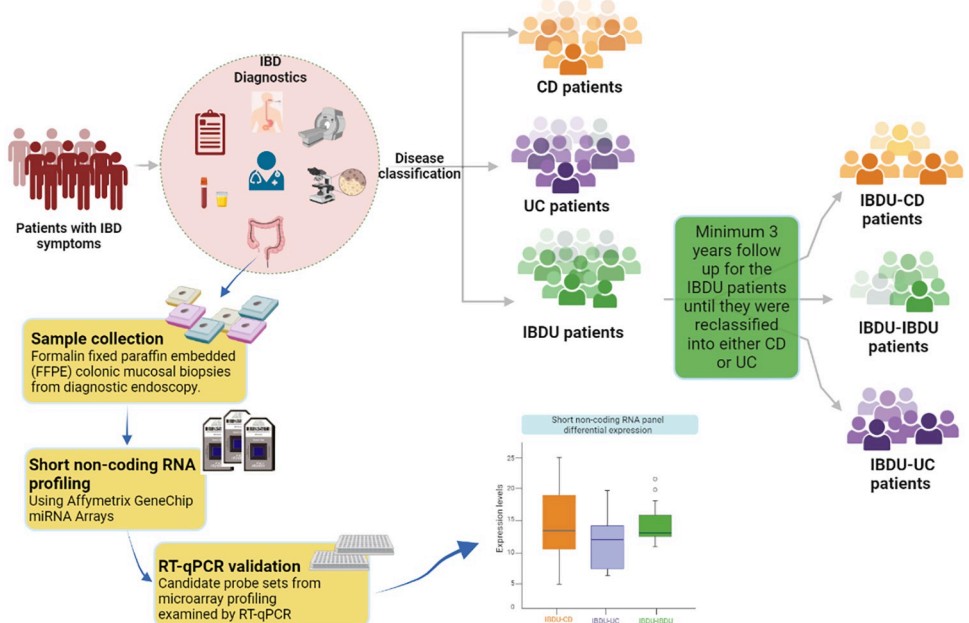

**Fig 1. Traditional IBD diagnosis and classification.** Patient cohort selection and workflow of the study are highlighted in yellow boxes.

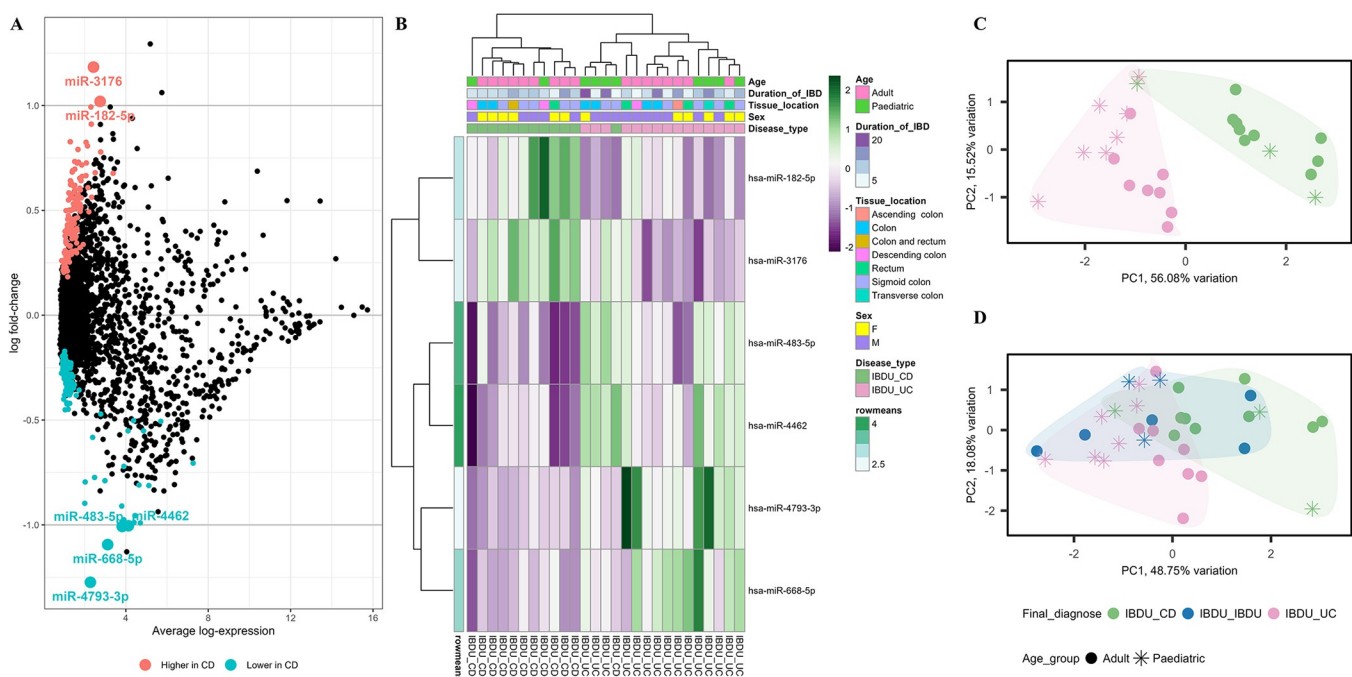

**Fig 2. Differential probe set expression analysis between IBDU-CD and IBDU-UC samples. (A)** A summary plot showing probe sets highly expressed or lowly expressed in IBDU-CD compared to IBDU-UC represented as blue and red dots, respectively. Probe sets with -1> log FC <1 is highlighted as larger dots and respective names are indicated. **(B)** Heatmap showing microarray-based expression of 6 probe sets. Age group, duration of IBD, tissue location, sex and disease type are shown as colored bars at the top of the heatmap. Expression levels are normalized to row means. **(C)** PCA plot showing IBDU-CD and IBDU-UC clustered under unsupervised clustering based the expression levels of the 6 probe sets. **(D)** PCA plot with IBDU-CD, IBDU-UC and IBDU-IBDU samples based the expression levels of the 6 probe sets. Different symbols represent different age groups among the samples.

samples gathered into separate clusters in the principal component analysis (PCA) plot (Fig 2C) based on the expression of the six probe sets. When plotted with other samples, eight of the IBDU-IBDU samples showed a tendency to be either IBDU-CD, IBDU-UC, or IBDU-IBDU (Fig 2D). Within the IBDU-UC cluster, samples were also grouped as adult or paediatric, indicating the need for additional analysis based on age at diagnosis.

In adult samples (n = 22), differential expression analysis revealed 499 probe sets with altered expression (adjusted P value <0.05) between IBDU-CD and IBDU-UC (Fig 3A and 3B). Among the differentially expressed probe sets, 10 of them (miR-1298-3p, x st ENSG00000239080, st ENSG00000239080 (snoU13), miR-451a, miR-628-5p, miR-4793-3p, miR-1273d, miR-3619-5p, U15B and hsa-miR-548x-3p) had a log fold change of -1<log FC>2 and were selected for further analysis. Unsupervised sample clustering based on the expression levels of the 10 probe sets revealed clustering of IBDU-CD and IBDU-UC samples (Fig 3C). The IBDU-IBDU samples showed a tendency towards either being in the IBDU-CD or IBDU-UC clusters (Fig 3D).

**IBDU samples vs definite CD and UC diagnosed samples.** In the microarray dataset, we identified a list of probe sets that were differentially expressed between IBDU-CD and IBDU-UC samples from analyses 1 (with all samples) and 2 (with only adult samples); see also Table 2. Probe sets for miR-3176, miR-182-5p, miR-451a, miR-1298-3p, miR-548x-3p, and snoU13 had AUCs of above 80% in predicting IBDU-CD and IBDU-UC. Samples from patients who had been diagnosed as CD or UC at the time of disease onset were also evaluated using probe set candidates derived from analyses 1 and 2. The best candidate probe sets for differentiating IBDU-CD and IBDU-UC samples had relatively low AUC values in CD and UC sample discrimination. As example, miR-3176 had an AUC of 85.6% for differentiating

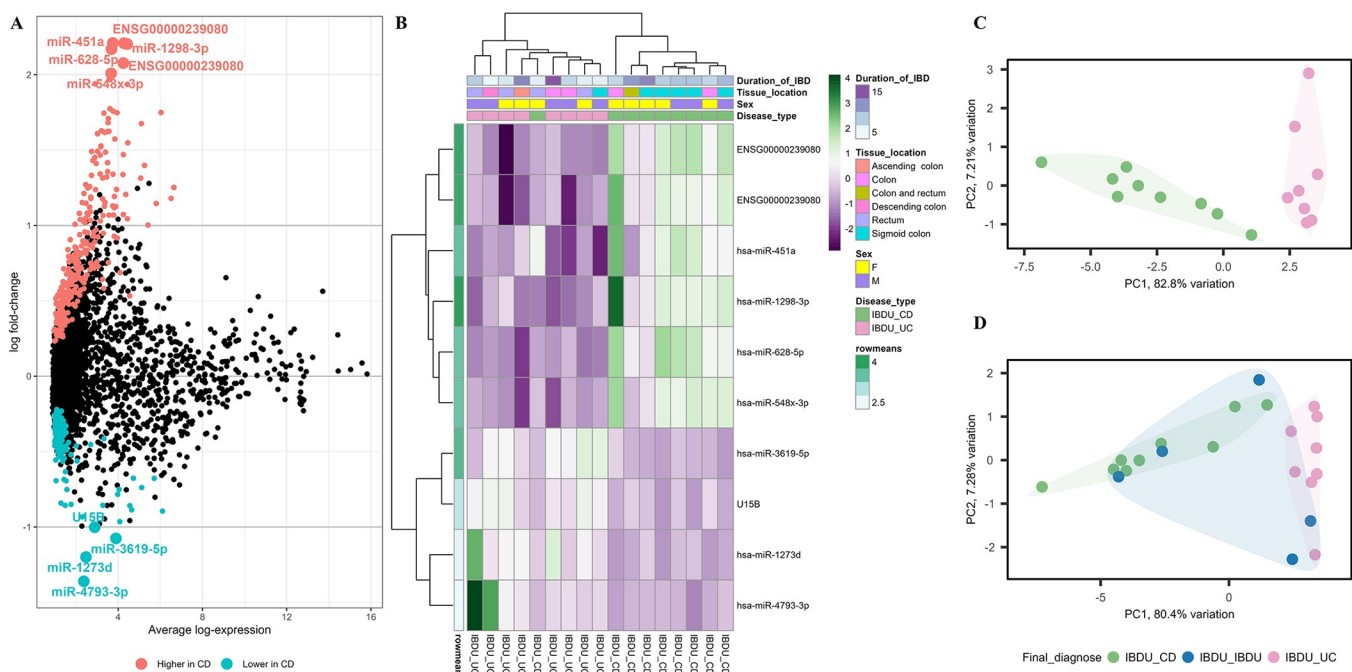

**Fig 3. Differential probe set expression analysis between adult IBDU-CD and IBDU-UC samples. (A)** Summary plot showing probe sets highly expressed or lowly expressed in IBDU-CD adults compared to IBDU-UC adults represented as blue and red dots respectively. Probe sets with -1> log FC <2 is highlighted as larger dots and respective names are indicated. **(B)** Heatmap showing microarray-based expression of the 10 probe sets. Duration of IBD, tissue location, sex and disease type are shown as colored bars at the top of the heatmap. Expression levels are normalized to row means. **(C)** PCA plot showing IBDU-CD and IBDU-UC clustered under unsupervised clustering based the expression levels of the 10 probe sets. **(D)** PCA plot with IBDU-CD, IBDU-UC and IBDU-IBDU samples based the expression levels of the 10 probe sets.

**Table 2. Results of analysis 1 and 2 based on microarray expression values of differentially expressed probe sets between IBDU-CD and IBDU-UC samples.**

| Analysis 1: Among all samples, $P < 0.05$ and $1 < \log FC < -1$ | | | |
|---|---|---|---|
| | **Probe set** | **IBDU-CD vs IBDU-UC**<br>**AUC (95% CI)** | **CD vs UC**<br>**AUC (95% CI)** |
| **Higher in CD** | miR-3176 | 85.6 (68.9–100) | 68.8 (25.2–100) |
| | miR-182-5p* | 81.7 (63.4–100) | 68.8 (20.8–100) |
| **Lower in CD** | miR-483-5p* | 71.7 (49.3–94) | 50 (0–100) |
| | miR-668-5p | 72.8 (50.5–95) | 62.5 (13.5–100) |
| | miR-4793-3p | 67.8 (45.7–89.8) | 62.5 (13.5–100) |
| | miR-4462 | 66.4 (44.1–88.7) | 43.8 (0–91.7) |
| **Analysis 2: Among only adults, $P < 0.05$ and $2 < \log FC < -1$** | | | |
| **Higher in CD** | miR-451a* | 84.4 (67.4–100) | 50 (1–99) |
| | miR-1298-3p* | 81.1 (63.2–99) | 81.2 (42.5–100) |
| | miR-628-5p* | 79.2 (60–98.3) | 87.5 (59.2–100) |
| | x_st_ENSG00000239080** | 80 (61.2–98.8) | 87.5 (59.2–100) |
| | st_ENSG00000239080** | 83.9 (67.4–100) | 87.5 (59.2–100) |
| **Lower in CD** | miR-4793-3p | 67.8 (45.7–89.8) | 62.5 (13.5–100) |
| | miR-1273d | 66.4 (44.4–88.4) | 56.2 (8.3–100) |
| | U15B | 65.0 (43.2–86.8) | 75.0 (35.0–100) |
| | miR-3619-5p* | 68.3 (47–89.8) | 65.6 (17.9–100) |
| | miR-548x-3p | 85.6 (69.7–100) | 87.5 (59.2–100) |

*—MirGeneDB 2.1

**- SnoRNA

between IBDU-CD and IBDU-UC, but between definite CD and UC diagnoses, the AUC was 68.8%. Similar trends were observed for probe sets miR-182-5p and miR-451a. Probe sets for miR-1298-3p, miR-548x-3p, and snoU13 showed AUCs above 80.0% in differentiating both IBDU-CD vs IBDU-UC and CD vs UC, when considered as individual probe sets (Table 2).

## Validation study

After filtering for MirGeneDB 2.1 database (miR-182-5p, miR-451a, miR-628-5p, miR-1298-3p, miR-4793-3p and snoU13), six sncRNAs from analysis 1 and 2 were selected for further RT-qPCR based validation in a subset of 32 samples used in the discovery study. Relative expression of snoU13 was found to be significantly higher in adult IBDU-CD samples compared to adult IBDU-UC samples (median, IBDU-CD: 0.037 and IBDU-UC: 0.006, $P = 0.028$, Fig 4A), thus, in agreement with their relative expression levels in the microarray analysis. Expression levels of miR-182-5p, miR-451a, and snoU13 in IBDU-CD and IBDU-UC samples are shown in Fig 4B. Based on the relative expression values of miR-182-5p, miR-451a, and snoU13 along with the age at diagnosis and sex of the patients, a generalized linear modelling (GLM) model was created and tested on a set of 7 IBDU-IBDU cases. The AUC value for discriminating IBDU-CD and IBDU-UC using the GLM model was 78.57% (95% CI: 60–97) (Fig 5). Expression of miR-628-5p, miR-1298-3p, and miR-4793-3p were found to be below the detection limit of the RT-qPCR method and excluded from the analysis.

## Discussion

IBDU can be perceived as a cumbersome diagnosis that can leave the patient in a limbo with an unsettled diagnosis that may turn into CD or UC. It also leaves the clinician with a significant

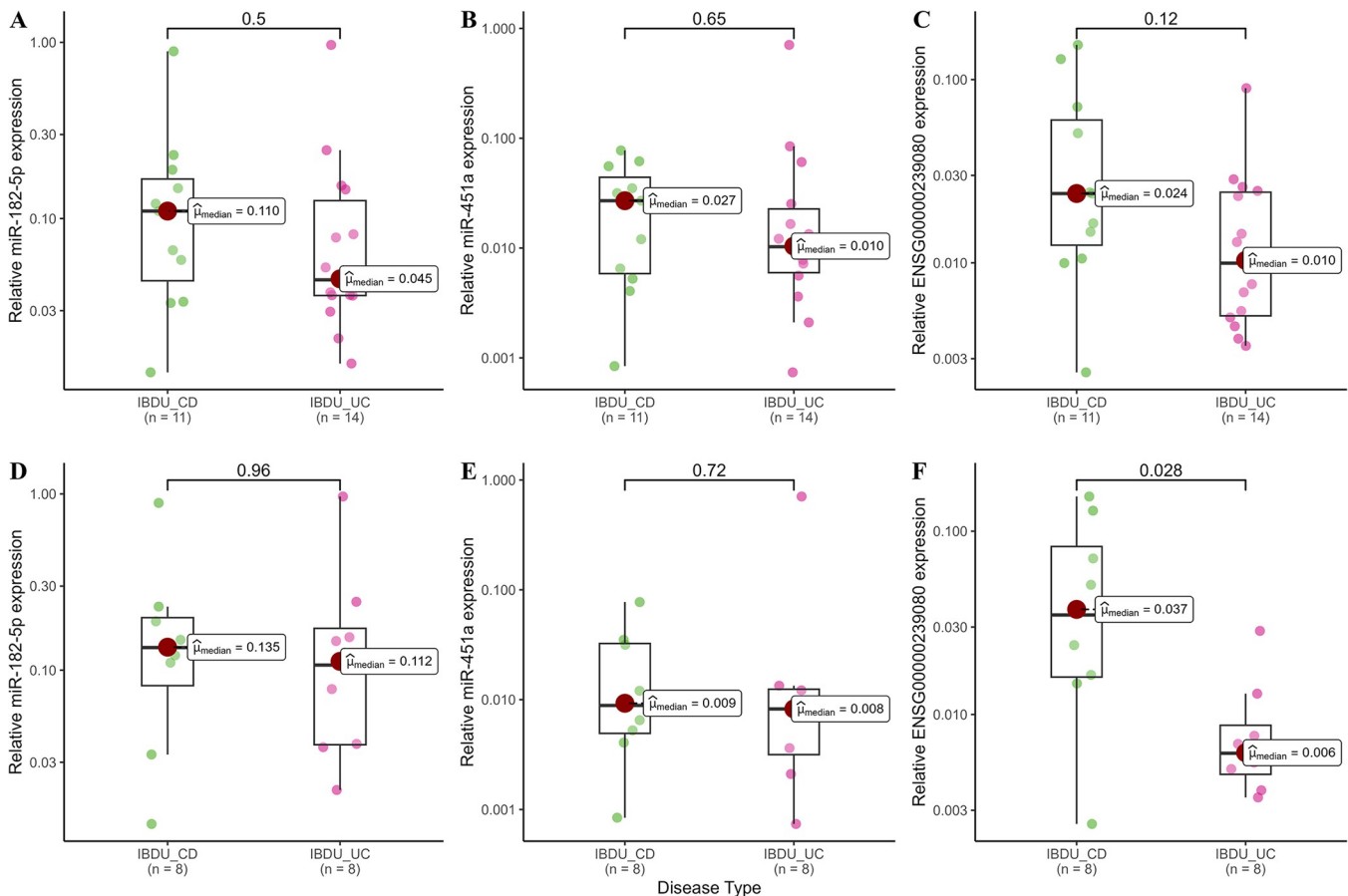

**Fig 4. Expression levels determined by RT-qPCR.** Box plots showing RT-qPCR based relative expression levels of miR-182-5p, miR-451a, and ENSG00000239080 (SnoU13) in all IBDU-CD and IBDU-UC samples **(A, B and C)** and among only adult IBDU-CD and IBDU-UC samples **(D, E and F)**.

tentative therapeutic strategy. Therefore, molecular biomarkers that, could help predicting a definite diagnosis at disease onset are of significant interest. Many studies on IBD have reported molecular profiles that can differentiate CD from UC [10,45]. However, those profiles have failed when applied to predicting CD or UC among IBDU cases [46–48], probably because the studies were based on definite CD and UC cases. In this study, we based our biomarker discovery analysis on IBDU patients with well-annotated follow-up. In our discovery analysis, we found several sncRNA transcripts with high prediction rates some of which were validated in RT-qPCR analyses. This resulted in an RT-qPCR-based three-gene-profile composed of miR-182-5p, miR-451a, and snoU13, together with the age and sex of the patient with an AUC value of 79%. In a clinical setting, the three-gene RT-qPCR-based GLM model, would thus help to sub-classify almost 4 out of 5 IBD patients diagnosed with IBDU, who would then receive the appropriate treatment based on their correct diagnosis avoiding years of delay.

We included pinch biopsy material from IBDU patients who were followed for a minimum of three years and up to 23 years. 77% of them were reclassified as CD or UC during the follow-up period. It has been estimated that 80% of IBDU cases are reclassified as CD or UC within 8 years of disease course [4,49], thus, our IBDU patient cohort appears representative for IBDU cases. Considering that reclassification occurred for 77% of the patients also indicates that the remaining 23% may still encounter reclassification. It has been shown that a majority will remain as IBDU through their lifespan [4,50].

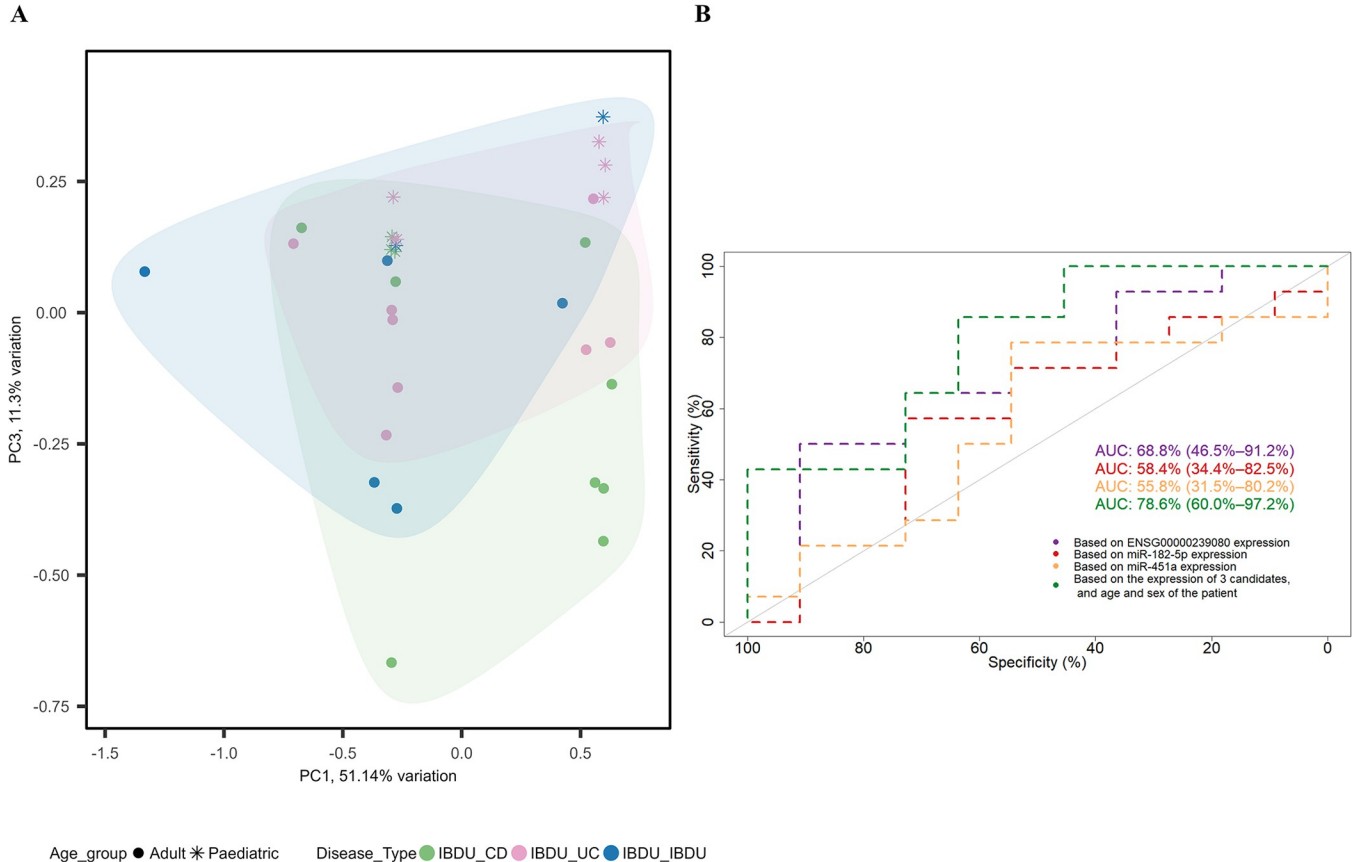

**Fig 5. A model based on the short noncoding RNA's expression, and the age and sex of the patient for prediction of CD and UC development from IBDU. (A)** PCA plot showing samples clustered based on the model into IBDU-CD (green) and IBDU-UC (pink). IBDU-IBDU samples shown in blue color show a tendency towards IBDU-CD or IBDU-UC. **(B)** ROC plot showing prediction based on 3 individual candidate's expression and the model based on the relative expression values of miR-182-5p, miR-451a, and ENSG00000239080 (SnoU13) along with the age at diagnosis and sex of the patients.

The sncRNAs identified to have significant predictive value in IBDU samples were also tested for their ability to discriminate CD from UC. In the microarray-based analysis, among the interesting candidates, miRNAs miR-1298-3p and miR-548x-3p, and the snoRNA snoU13 showed high discriminatory AUC values among adult IBDU cases, and a high prediction rate in discrimination of CD from UC. However, only the snoRNA could be detected and validated using RT-qPCR. In our three-gene RT-qPCR-based GLM model, we included miR-182-5p and miR-451a, which were detectable by RT-qPCR, and that showed predictive AUC values > 80% in the microarray analysis. As mentioned above, the GLM model based on the three gene-targets, resulted in an AUC of 79% in the prediction of IBDUs to turn into CD or UC, however, the GLM model failed to discriminate definite diagnosis of CDs from that of UCs. Our data suggest that dynamic changes of the molecular profiles of the CD and UC patients during the development of their disease may have affected the output of our pre-determined expression profile. On the other hand, the IBDU early disease stage may express a distinct molecular profile that is linked to only the IBDU diagnosis. We hypothesize that the molecular signatures of CD and UC can evolve over time among patients initially diagnosed with IBDU, and thus may exhibit distinct gene expression profiles at different stages during the disease course. Indeed, we noted that IBDU samples with no apparent phenotype of IBDU-CD or IBDU-UC have a distinct molecular profile different from IBD cases with

definite phenotypes of CD or UC, thus, supporting the assumption that the IBDU diagnosis comprise a particular entity different from IBDU-CD and IBDU-UC, here called IBDU-IBDU.

In the microarray analysis we found that the AUC values for each individual candidate probe set such as miR-182-5p, miR-451a and snoU13 were having AUC values above 80% in discriminating IBDU-CD from IBDU-UC, whereas the individual AUC values based on RT-qPCR expression values were reduced to below 70%. A possible explanation for this could be that in the validation using RT-qPCR method the expression values are normalized to the geometrical mean of a set of 5 reference candidates to reduce variation related to the technique.

Earlier studies have shown an age dependent variation in phenotypes for several diseases including IBD [51–53]. In their review, Ruel et al. stated that the characteristics and natural history of IBD appear to vary depending on the age of onset [54]. Also, phenotypic expression profiles of late-onset IBD compared with those in the younger population were found to be different [55,56]. Therefore, the age dependent clustering of samples in our discovery study was not surprising. Age dependent expression differences of miRNAs and snoRNAs have also been reported by others [57,58].

## Biological aspects of sncRNA target candidates

IBD is an inflammatory condition that is believed to be caused by compromised barrier function, where the epithelial barrier is penetrated by bacterial influx, and un-balanced immune response that insufficiently fight the infection. Generally, CD is characterized by inflammation throughout the gastrointestinal tract and by transmural inflammation, whereas UC is characterized by relapsing and remitting mucosal inflammation extending from the rectum to proximal segments of the colon. However, other parameters also contribute to the characteristics of the two diseases. Changes in molecular expression in IBD may be related to one or more of these processes. Based on our microarray data, the three target genes, miR-182-5p, miR-451a, and snoU13, in our GLM model, were all higher in IBDU-CD compared to IBDU-UC.

MiR-182-5p is one of three miRNAs in the miR-183/96/182 cluster. miR-182 has been reported to be strongly induced in B cells [59]. In miR-182 deficient mice, miR-182 was found to be involved in antibody response at early time-points during immunization when challenged with a T-cell-dependent antigen and suggested to be responsible for defective extrafollicular response [60]. In DSS-induced UC in mice, inhibition of miR-182-5p was reported to ameliorate intestinal barrier dysfunction [61]. One of the interesting targets of miR-182-5p in the relation to IBD, is Claudin-2, which is one of the tight junction proteins downregulated in inflammatory bowel disease [62]. Taken together, elevated miR-182-5p levels in IBD may contribute to both an unbalanced immune response and to disrupted epithelial barrier function.

MiR-451a has been shown to serve as a blood-derived biomarker in a variety of conditions, including colorectal cancer (CRC) and arthritis [63,64]. In the study by Juzenas et al., [65] of miRNAs in human peripheral blood, miR-451a was found most abundant in CD235a positive cells, which are considered erythrocytes, suggesting that the miR-451 levels may be directly related to bleeding or inflammation. Moret-Tatay et al. developed a panel of plasma miRNAs that included miR-451a to identify patients at high post-operative recurrence risk among CD [66]. In a study of bone marrow derived macrophages from miR-451 knock-out mice, it was reported that miR-451 and reactive oxygen species (ROS) functionally crosstalk to regulate macrophage oxidant stress [67]. Our finding that miR-451a is higher in IBDU-CD than IBDU-UC may reflect different inflammatory stages of the disease in terms of presence of erythrocytes, ROS, and macrophages.

The third candidate in our panel is the snoRNA with the Ensembl gene identifier ENSG00000239080 (snoU13), which is also known as LOC124905271 or small nucleolar RNA

U13 [68]. SnoU13 is poorly described in the literature, but has been found to contribute to biomarker profiles [69,70]. Yang et al. found that the expression of snoRA15, snoRA41 and snoRD33 was upregulated in UC and in CRC tissues compared with matched non-cancerous tissues, with an increasing trend from healthy control to UC and CRC [27]. Exploring the functions of snoRNAs in IBD will be needed to better understand their role.

The presented study is exploratory with a limited number of patient cases. The unique sample set with numerous inclusion criteria limits the number of samples that were accessible to investigate in this study. However, the IBDU cases included were carefully followed over time and allowed identification of differentially expressed sncRNA. The number of paediatric patients was particularly few, which prevented detailed analysis of the paediatric group. The remaining group of IBDU-IBDU patients may ultimately settle with a CD or UC diagnosis, however, this may still take years to occur. This of course directs our study towards the need for biomarkers that can helps reclassifying them to CD or UC or an independent group with unique characteristics that could be defined as IBDU group, and thereby aid in providing the optimal treatment for the patients.

## Conclusions

In summary, we have developed a prediction model based on the expression of three sncRNAs, and age and sex of the patient that can assist the clinician in properly classifying IBDU patients into CD or UC at disease onset. Our study also suggests that true IBDU patients may have different pathogenic mechanisms compared to IBDU patients later diagnosed with CD or UC. Further research based on a larger cohort of real scenario of IBD patients could help in understanding the distinct molecular pathways of IBDU subgroups. Different factors such as age, gender, and other disease characteristics should be thoroughly investigated in a larger patient cohort. Future developments on diagnostic strategies and personalized therapies for IBDU and possible development towards CD or UC are necessary, which will plausibly introduce biomarkers such as miRNAs and snoRNAs as promising components to the analysis tool kit.

## Supporting information

**S1 File. The primer sequences for all the short non-coding RNAs used in the RT-qPCR validation.**
(PDF)

## Acknowledgments

We would like to thank laboratory technicians Christina Grønhøj, Department of pathology, Herlev hospital and Trine Møller, Bioneer A/S for their valuable technical assistance.

## Author Contributions

**Conceptualization:** Jaslin P. James, Lene Buhl Riis, Mikkel Malham, Estrid Høgdall, Ebbe Langholz, Boye Schnack Nielsen.

**Data curation:** Jaslin P. James.

**Formal analysis:** Jaslin P. James.

**Funding acquisition:** Jaslin P. James, Lene Buhl Riis.

**Investigation:** Jaslin P. James, Rolf Søkilde.

**Methodology:** Jaslin P. James, Rolf Søkilde, Boye Schnack Nielsen.

**Project administration:** Jaslin P. James, Lene Buhl Riis, Estrid Høgdall.

**Resources:** Jaslin P. James, Mikkel Malham, Ebbe Langholz, Boye Schnack Nielsen.

**Software:** Jaslin P. James, Rolf Søkilde.

**Supervision:** Lene Buhl Riis, Mikkel Malham, Estrid Høgdall, Ebbe Langholz, Boye Schnack Nielsen.

**Validation:** Jaslin P. James, Rolf Søkilde.

**Visualization:** Jaslin P. James.

**Writing – original draft:** Jaslin P. James.

**Writing – review & editing:** Lene Buhl Riis, Rolf Søkilde, Mikkel Malham, Estrid Høgdall, Ebbe Langholz, Boye Schnack Nielsen.

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
