## [Decision Letter · Decision Letter 0]

10 Oct 2023

PONE-D-23-17042Short noncoding RNAs as predictive biomarkers for the development from inflammatory bowel disease unclassified to Crohn’s disease or ulcerative colitisPLOS ONE

Dear Dr. James,

Thank you for submitting your manuscript to PLOS ONE. After careful consideration, we feel that it has merit but does not fully meet PLOS ONE’s publication criteria as it currently stands. Therefore, we invite you to submit a revised version of the manuscript that addresses the points raised during the review process.

We look forward to receiving your revised manuscript.

Kind regards,

Kira Astakhova

Academic Editor

PLOS ONE

Journal Requirements:

Reviewers' comments:

Reviewer's Responses to Questions

**Comments to the Author**

1. Is the manuscript technically sound, and do the data support the conclusions?

Reviewer #1: Yes

Reviewer #2: Yes

2. Has the statistical analysis been performed appropriately and rigorously? 

Reviewer #1: Yes

Reviewer #2: Yes

3. Have the authors made all data underlying the findings in their manuscript fully available?

Reviewer #1: Yes

Reviewer #2: Yes

4. Is the manuscript presented in an intelligible fashion and written in standard English?

Reviewer #1: Yes

Reviewer #2: Yes

5. Review Comments to the Author

Reviewer #1: In this paper, James et al. present a novel molecular profile of the IBD subtypes of ulcerative colitis (UC) and Crohn’s disease (CD) and even suggest a potential third yet undescribed subtype. In this retrospective study, they analyzed the molecular profile of mucosal samples of adult and pediatric patients, then a subset of adult patients, previously diagnosed with unclassified IBD then through clinical follow up reclassified into unclassified, UC, or CD. They clearly explain their methodology for eliminating molecular targets. They show that a model using three probe sets, age, and sex resulted in an AUC of 78.6%. Considering that there are unique treatment options for UC and CD, this research is a step in the right direction for advancement of clinical care and is important work. The study design is well done and with the addition of suggested clarifications detailed below, I believe that the manuscript would be appropriate for the scope of this journal. Well done James et al..

Major points

1. The samples used were collected between 1998 - 2018. Despite spanning 10 years, only a small number of samples were available for analysis. Author should make clear the inclusion and exclusion criteria for this study. This is a critical part of the methodology to include as it may make it such that their results only apply to a small subset of the IBD population which could substantially alter the generalizability of their findings.

2. Authors should include in their Discussion potential reasons for the poor performance of candidate probe sets in discriminating between definite CD and UC diagnoses as compared to IBDU-CD and -UC samples.

Minor points

1. The authors should aim for consistency in terms. The term “short nc RNAs” and “short non-coding RNAs” are both used. For clarity, it would be helpful for the reader if the term was written in full in the introduction with the abbreviation introduced immediately after then referred to by the abbreviation in the remainder of the manuscript.

2. Line 159 - 160: Please expand briefly as to why the two probe sets were not included for further validation.

Other:

Line 178: I believe the authors meant diagnosis instead of diagnose.

Reviewer #2: The manuscript by James et al. investigates the application of short noncoding RNA profiles for early reclassification of IBDU patients into IBDU-CD or IBDU-UC. As the therapeutic methods of the disease subsets differ, early reclassification allows appropriate clinical management. The study applies a microarray study for probe selection, which is validated by RT-qPCR. miR-182-5p, miR-451a and snoU13 together with age and sex resulted in an AUC of 78.6% (95 % CI: 60 - 97) in discriminating IBDU-CD from IBDU-UC. I recommend accepting the manuscript after minor revision.

- For the figures in general, the text and axis values are difficult to read and should be enlarged. This especially applies to the boxplot in Figure 1, the text and values in Figure 2 and 3, as well as the color legends in Figure 5.

- For the manuscript full title, it is suggested to replace “development” with “differential diagnosis”, “reclassification” or similar, as it better reflects the applicability and aim of the study.

- Page 2, line 26. “Of 35 IBDU patients initially diagnosed with IBDU, 12, 15, and eight 27 were reclassified into CD(IBDU-CD), UC(IBDU-UC), or remained as IBDU(IBDU-IBDU) respectively.” When reading the abstract, it was not clear to me that this reclassification was not part of the study. It might be specified.

- Page 17, line 277: “Many studies on IBD have reported molecular profiles that can differentiate CD from UC. However, those profiles have failed when applied to predicting CD or UC among IBDU cases”. It is recommended to comment on the reason why these methods are failing.

6. PLOS authors have the option to publish the peer review history of their article (what does this mean?). If published, this will include your full peer review and any attached files.

Reviewer #1: No

Reviewer #2: No

---

## [Author Response · Author response to Decision Letter 0]

15 Nov 2023

Emily Chenette

Editor-in-Chief

PLOS ONE

Date: 10th November 2023

Dear Dr.Chenette,

Re: Resubmission of manuscript reference No. PONE-D-23-17042

Please find attached a revised version of our manuscript entitled “Short noncoding RNAs as predictive biomarkers for the development from inflammatory bowel disease unclassified to Crohn’s disease or ulcerative colitis”, which we would like to resubmit for consideration for publication in PLOS ONE. 

The reviewer’s comments were highly insightful and enabled us to greatly improve the quality of our manuscript. In the following pages are our point-by-point responses to each of the comments.

We are also attaching the revised manuscript with track changes. In accordance with the reviewers’ comments, more information on patient characteristics and more elaborated discussions are added to the revised manuscript. 

We hope that the revisions in the manuscript and our accompanying responses will be sufficient to make our manuscript suitable for publication in the journal. 

We thank you for your consideration,

Jaslin Pallikkunnath James

Department of Pathology, Herlev Hospital, Denmark

jaslin.pallikkunnath.james@regionh.dk

We thank the reviewers for their thorough review of our work and all the valuable comments. In the following we have addressed the comments point-to-point.

Reviewers' comments:

Reviewer #1:

In this paper, James et al. present a novel molecular profile of the IBD subtypes of ulcerative colitis (UC) and Crohn’s disease (CD) and even suggest a potential third yet undescribed subtype. In this retrospective study, they analyzed the molecular profile of mucosal samples of adult and pediatric patients, then a subset of adult patients, previously diagnosed with unclassified IBD then through clinical follow up reclassified into unclassified, UC, or CD. They clearly explain their methodology for eliminating molecular targets. They show that a model using three probe sets, age, and sex resulted in an AUC of 78.6%. Considering that there are unique treatment options for UC and CD, this research is a step in the right direction for advancement of clinical care and is important work. The study design is well done and with the addition of suggested clarifications detailed below, I believe that the manuscript would be appropriate for the scope of this journal. Well done James etal..

Major-points

1. The samples used were collected between 1998 - 2018. Despite spanning 10 years, only a small number of samples were available for analysis. Author should make clear the inclusion and exclusion criteria for this study. This is a critical part of the methodology to include as it may make it such that their results only apply to a small subset of the IBD population which could substantially alter the generalizability of their findings.

• We thank reviewer for the insightful point. We fully agree that the number of samples included in the study is relatively small. The unique sample cohort was the result of several inclusion criteria such as availability of biopsies at the time of diagnosis, minimum of 3 years as IBDU, and making sure that all patients had thorough clinical evaluation to reduce the risk of misclassification, which altogether limits the number of samples that were available to us. By using these strict criteria, we believe that they increased the chance of obtaining useful data. 

• We have clarified the inclusion and exclusion criteria in the samples section on page 7.

2. Authors should include in their Discussion potential reasons for the poor performance of candidate probe sets in discriminating between definite CD and UC diagnoses as compared to IBDU-CD and -UC samples.

• We appreciate the insightful recommendation. The molecular signatures of CD and UC can evolve over time among patients initially diagnosed with IBDU, as well as the IBDU patients may exhibit distinct gene expression profiles at different stages during the disease's progression. These dynamic changes might go unnoticed by probe sets created using static profiles. Patients with an unsettled diagnosis (IBDU) can have a broad spectrum of clinical, histological, and genetic characteristics associated with both CD and UC. Therefore, patient biopsies taken at the time of diagnosis, when there were overlapping characteristics of both CD and UC, were used in this investigation. It was not possible to adapt the probesets, which in our study were created from a heterogeneous group of IBDU patients, to the comparatively more homogeneous definite CD/UC population.

• We have added these considerations to the manuscript (Page 19), lines 317-323

Minor points

1. The authors should aim for consistency in terms. The term “short nc RNAs” and “short non-coding RNAs” are both used. For clarity, it would be helpful for the reader if the term was written in full in the introduction with the abbreviation introduced immediately after then referred to by the abbreviation in the remainder of the manuscript.

• We have clarified this issue throughout the manuscript

2. Line 159 - 160: Please expand briefly as to why the two probe sets were not included for further validation.

• Thank you for pointing this out. Please take a note that candidates miR-4793-3p and ENSG00000239080 (snoU13) were indeed included in the RT-qPCR validation. 

• In lines 166-168, we have clarified this misunderstanding.

Other:

Line 178: I believe the authors meant diagnosis instead of diagnose.

• Thank you for pointing out this. We have made the correction in line 186 in the revised manuscript.

 

Reviewer #2:

The manuscript by James et al. investigates the application of short noncoding RNA profiles for early reclassification of IBDU patients into IBDU-CD or IBDU-UC. As the therapeutic methods of the disease subsets differ, early reclassification allows appropriate clinical management. The study applies a microarray study for probe selection, which is validated by RT-qPCR. miR-182-5p, miR-451a and snoU13 together with age and sex resulted in an AUC of 78.6% (95 % CI: 60 - 97) in discriminating IBDU-CD from IBDU-UC. I recommend accepting the manuscript after minor revision.

- For the figures in general, the text and axis values are difficult to read and should be enlarged. This especially applies to the boxplot in Figure 1, the text and values in Figure 2 and 3, as well as the color legends in Figure 5.

• Thank you for this suggestion. We ensured that all images and text elements were legible and clear. It's possible that the PDF format for the manuscript submission may affect the way figures appear when viewed online, but the original figures were of high quality.

- For the manuscript full title, it is suggested to replace “development” with “differential diagnosis”, “reclassification” or similar, as it better reflects the applicability and aim of the study.

• Thank you for this suggestion. After thorough discussion and consultation among the authors, we believe that the title effectively conveys the message of the research and maintain a relevant and sufficient level of clarity. 

- Page 2, line 26. “Of 35 IBDU patients initially diagnosed with IBDU, 12, 15, and eight 27 were reclassified into CD(IBDU-CD), UC(IBDU-UC), or remained as IBDU(IBDU-IBDU) respectively.” When reading the abstract, it was not clear to me that this reclassification was not part of the study. It might be specified.

• Thank you for pointing out this important aspect. We have revised the line 26 in the abstract on Page 2. 

- Page 17, line 277: “Many studies on IBD have reported molecular profiles that can differentiate CD from UC. However, those profiles have failed when applied to predicting CD or UC among IBDU cases”. It is recommended to comment on the reason why these methods are failing. 

• Thank you for the valuable suggestion. The majority of the prior studies were developed using distinct CD and UC cases. Patients who are initially diagnosed with IBDU may have changing molecular markers of CD and UC throughout time, and they may also show diverse gene expression profiles at different points in the disease's course. Furthermore, IBDU is frequently a less well-defined category, and the paucity of data for this particular group may have impeded the development of reliable molecular predictive models in the past.

• We have added this information to page 18, lines 291-292.

---

## [Decision Letter · Decision Letter 1]

3 Jan 2024

Short noncoding RNAs as predictive biomarkers for the development from inflammatory bowel disease unclassified to Crohn’s disease or ulcerative colitis

PONE-D-23-17042R1

Dear Dr. James,

We’re pleased to inform you that your manuscript has been judged scientifically suitable for publication and will be formally accepted for publication once it meets all outstanding technical requirements.

Kind regards,

Kira Astakhova

Academic Editor

PLOS ONE
---

## [Editor Report · Acceptance letter]

15 Feb 2024

PONE-D-23-17042R1 

PLOS ONE

Dear Dr. James, 

I'm pleased to inform you that your manuscript has been deemed suitable for publication in PLOS ONE. Congratulations! Your manuscript is now being handed over to our production team.

Kind regards, 

on behalf of

Dr. Kira Astakhova 

Academic Editor

PLOS ONE